# A Method of Bus Network Optimization Based on Complex Network and Beidou Vehicle Location

**Peixin Dong [1] , Dongyuan Li [1], Jianping Xing [1,\*], Haohui Duan [2] and Yong Wu [3]**

[1] School of Microelectronics, Shandong University, Jinan 250100, China; dongpeixin1995@163.com (P.D.); 17862978251@163.com (D.L.)

[2] Traffic Police Detachment, Jinan Public Security Bureau, Jinan 250100, China; 15288840081@163.com

[3] Jinan Public Transportation Corporation, Jinan 250100, China; ahlangzai@163.com

\* Correspondence: xingjp@sdu.edu.cn; Tel.: +86-185-5310-1316

**Abstract:** Aiming at the problems of poor time performance and accuracy in bus stops network optimization, this paper proposes an algorithm based on complex network and graph theory and Beidou Vehicle Location to measure the importance of bus stops. This method narrows the scope of points and edges to be optimized and is applied to the Jinan bus stop network. In this method, the bus driving efficiency, which can objectively reflect actual road conditions, is taken as the weight of the connecting edges in the network, and the network is optimized through the network efficiency. The experimental results show that, compared with the original network, the optimized network time performance is good and the optimized network bus driving efficiency is improved.

**Keywords:** Beidou Vehicle Location; bus stops network; network efficiency; optimize

---

## 1. Introduction

As an important component of the infrastructure and transport system, urban public transportation is directly related to the economic development of the city and people's livelihood, with a comprehensive and guiding impact on the urban economy. Giving priority to the development of urban public transportation is an important means to improve the utilization efficiency of traffic resources and alleviate traffic congestion, which is essential to solving the traffic congestion problem. Reasonable analysis of the characteristics of the public transport network functions effectively in improving the travel efficiency of urban residents. Although the research into urban bus network systems is deepening, there still exist some problems with respect to poor time performance and poor accuracy, so in the analysis of urban public transport networks, it is becoming increasingly urgent to develop a more accurate and comprehensive calculation method.

In [1], a network efficiency calculation method is proposed, but this method is general, and is applicable to simple topological structure networks or different weight networks. At the same time, with the continuous development of complex network theory research, by combining this theory with public transport networks, the urban public transport network system can be abstracted into a complex network system, which is of great significance for the study of the structural characteristics of real public transport networks and the optimization of public transport networks. In terms of basic models of complex networks and public transport networks, Erdos et al. [2] found and proposed ER models of stochastic networks that are different from regular networks. Watts et al. [3] mentions and establishes a small-world network model. The research results show that many real-world networks, especially large-scale networks, have small-world characteristics; that is, the average distance in networks is small, and the clustering coefficient is large. With regard to the weighted networks of urban public transport, in combination with the application of research into complex networks in real life, a weighted

network is proposed in [4,5], so as to classify the network topology and community. In this paper, an improved calculation method of network efficiency is proposed on the basis of factors such as passenger carrying rate, and a new optimization method for public transport networks is proposed based on complex networks.

## 2. Graph Theory and Complex Network of Public Transportation

The graph is composed of set V and set E, denoted as $G = (V, E)$. Set V is a set of vertices of a graph, which are generally used to represent individuals in a real system; set E is a set of edges of a graph, which is a set of relations between vertices, most of which are used to represent the relationships or interactions among individuals in a real system.

In $G = (V, E)$, if any node for $a, b \in V$, when $(a, b) \in E$, $(b, a) \in E$ may not be true, then we call this graph a directed graph. In the directed graph, two related nodes are usually connected by the edges with an arrow; when $(a, b) \in V$, $(b, a) \in E$, then we call this this graph an undirected graph. The edge of an undirected graph has no direction, and the relation between vertices is represented by an edge without an arrow. The shortest path in an undirected and unweighted graph is the path with the smallest total distance from the source node to the destination node among all possible paths, while the shortest path of an undirected and weighted graph is that with the smallest sum of weights.

This paper uses graph theory and the basic theory of complex networks to model and analyze a city bus station network [6], taking the station as the node, and with edges existing between two adjacent stations belonging to the same bus line. The bus network studied in this paper is an undirected graph, which does not distinguish between uplink and downlink, and uplink and downlink symmetric stations are considered to be the same station [7].

The adjacency matrix A (n * n) of the bus network is used to express the connection relationship between bus stations. n represents the number of stations in the network, $a_{ij} = 1$ indicates that the station *i* and station *j* belong to the same bus route and are adjacent; $a_{ij} = 0$ represents that station *i* and station *j* are not adjacent or do not belong to the same line. Figure 1 is an adjacency matrix of a line in a bus station network. This matrix uses 0,1 to represent the connection relationship of each station. Based on this matrix, the complex bus network diagram of Jinan can be modeled as shown in Figure 2.

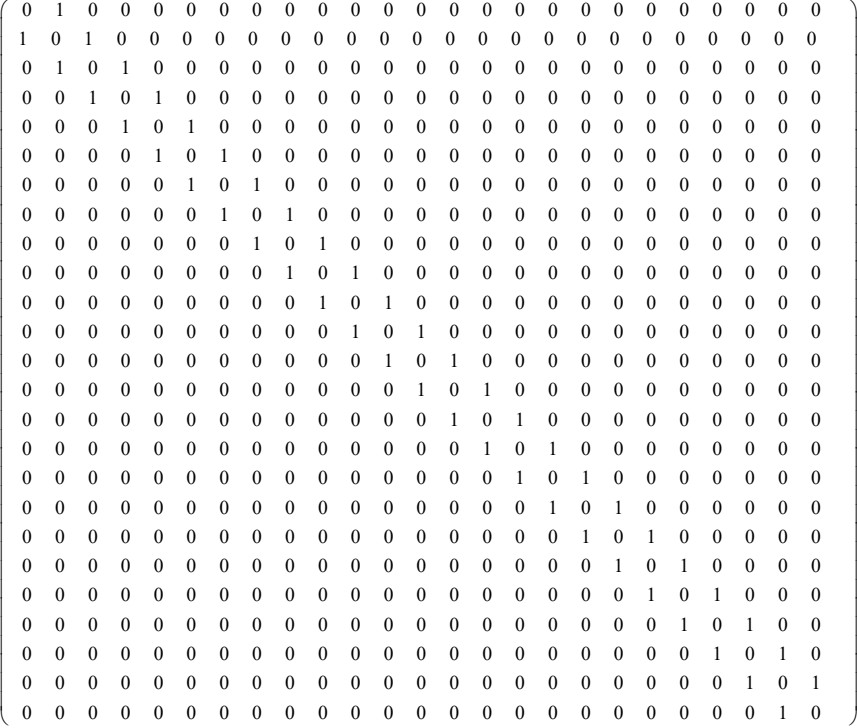

**Figure 1.** Adjacency matrix of a bus station network.

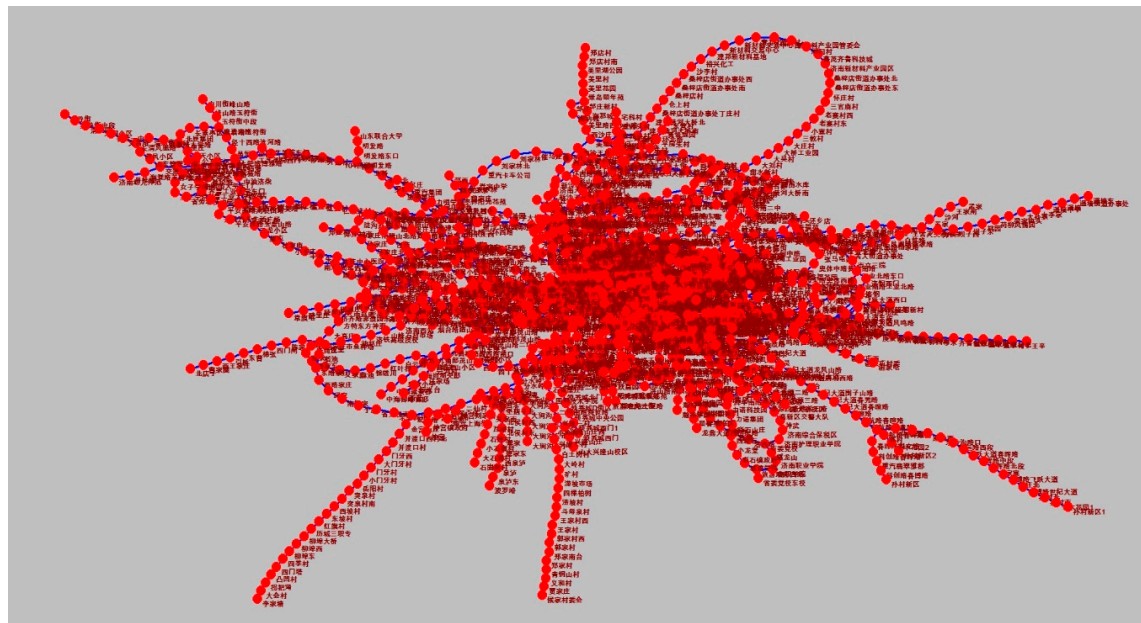

**Figure 2.** Public transportation network system floor plan. (The Chinese in the figure is the actual place name of Jinan City.).

## 3. Weight Definition Method of Weighted Bus Network

In the above network, the weight of the edge is 0 or 1, which only indicates the connection relationship between the stations in the bus station network [8]. The weighted network is expressed through any edge of any two connected nodes having a particular weight in the complex network [9]. In the weighted bus network, there is a weight for the edge of any two adjacent stations *i* and *j*, which is generally expressed by the bus driving efficiency.

Compared with the unweighted bus network [10], the weighted bus network takes into account a variety of factors, such as distance, road carrying capacity, passenger load rate, passenger flow, waiting time of passengers, etc., which can reflect the driving situation of the bus in different sections more objectively, comprehensively and truly. In this paper, the bus driving efficiency is used as the edge weight [11,12].

### 3.1. Road Impedance Function

Road impedance is a physical quantity used to indicate the magnitude of resistance encountered by a vehicle on the road, reflecting the fluency of vehicle passing through a road section. Because the vehicle travel time on the road can directly reflect the degree of obstruction in the process of vehicle driving, the vehicle travel time is generally used to describe the road impedance.

BPR (Bureau of Public Roads) function is usually introduced when calculating the travel time of public transport vehicles. The BPR function is as follows:

$$T_z = T_z^0 \left[ 1 + \alpha \left( \frac{x_z}{c_z} \right)^\beta \right] \tag{1}$$

$T_z^0$ represents the impedance value of the road section $z$ when the traffic flow is 0; $x_z$ is the traffic flow of road section $z$; $c_z$ represents the actual capacity of section $z$; $T_z$ is the vehicle travel time of the section $z$; $\alpha$ and $\beta$ are correction coefficients, the US Highway Board recommends $\alpha = 0.15$, $\beta = 4$. Thus, the bus travel time between the two adjacent stations is obtained:

$$t_{ij} = \sum_{z \in S_{ij}} P_{ijz} T_z \tag{2}$$

$S_{ij}$ represents all the sections of the road; $P_{ijz}$ represents the proportion of passing distance to the total distance of section $z$. The distance of a road section is acquired by Beidou high-precision vehicle positioning equipment in real time.

### 3.2. Calculation Method of Bus Driving Efficiency

The physical meaning of bus efficiency refers to the number of passengers served by the unit bus impedance; that is, the bus efficiency between stations is proportional to the bus passenger flow between stations, and is inversely proportional to the impedance between stations, and the impedance can be expressed as a temporal form. Therefore, the calculation method of bus driving efficiency can be put forward:

$$W_{ij} = \frac{x_{ij}}{t_{ij}} \tag{3}$$

$W_{ij}$ represents the bus driving efficiency between station $i$ and station $j$; $x_{ij}$ represents the passenger flow between station $i$ and station $j$, which can be obtained by the OD matrix obtained from the IC card swipe data on the bus; $t_{ij}$ represents the bus travel time between station $i$ and station $j$, which can be obtained by BPR function. $W_{ij}$ is the weight of the edge, the greater the weight of the edge, the higher the efficiency of the bus running on the road [13].

In Equation (3), the bus driving efficiency is proportional to the passenger flow, and the bus travel efficiency will increase linearly with the increase of passenger flow. However, in real life, when the passenger flow increases beyond a certain extent, it may lead to traffic congestion and reduce bus driving efficiency, so this formula does not accord with common sense and needs to be improved.

### 3.3. Improved Calculation Method for Bus Driving Efficiency

In actual situations, when we calculate passengers' time from station $i$ to station $j$, in addition to the bus travel time, we should also consider the waiting time of passengers [14]. The waiting time of passengers is related to the number of bus routes, but also to the passenger flow $x_{ij}$; when a bus line tends to be fully loaded, the probability of passengers missing the bus is greater, resulting in more waiting time. Therefore, an improved calculation method for bus driving efficiency is put forward:

$$W_{ij} = \frac{x_{ij}}{t_{ij}\left(1 + \lambda\left(\frac{x_{ij}}{x_{max}}\right)^{\omega}\right)} \tag{4}$$

$x_{max}$ is the upper limit for bus passenger volume; $\omega$ and $\lambda$ are correction coefficients; according to the actual situation of bus line operation in the city of Jinan, this paper set up $\lambda = 1$, $\omega = 10$. The improved $W_{ij}$ is used as the weight of the edge.

The improved results are shown in Figure 3. As can be seen from the figure, in the traditional algorithm, the bus driving efficiency linearly increases with the increase of passenger load rate; however, in the improved algorithm, after adjusting passenger volume and other factors, a clear line of demarcation could be seen: when the passenger load rate is below 80%, the bus driving efficiency will increase linearly with the increase of passenger load rate; when the passenger load rate reaches 80%, the bus driving efficiency will decrease gradually with the increase of passenger load rate.

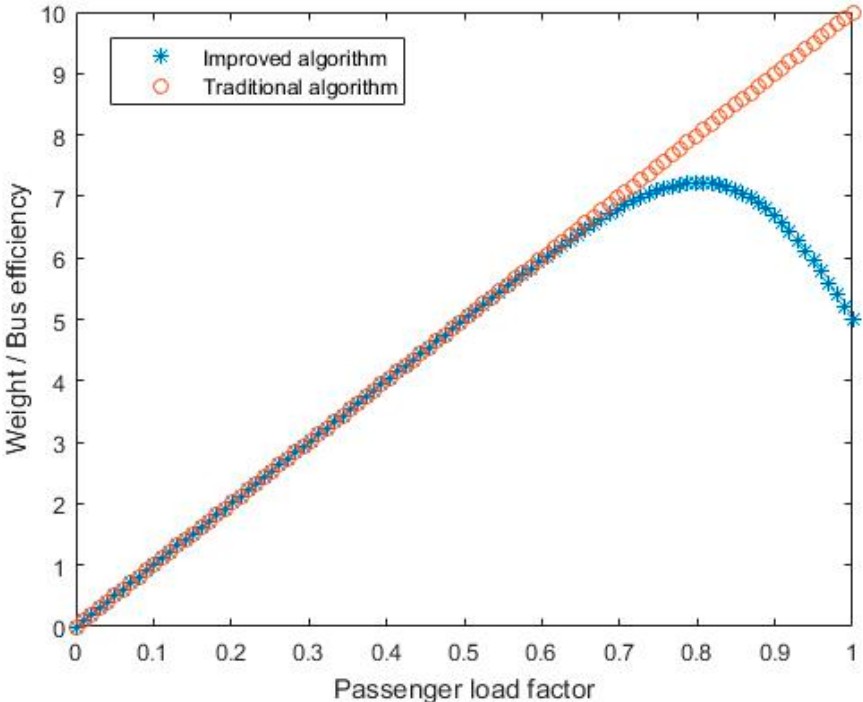

**Figure 3.** Two kinds of bus efficiency.

*3.4. Construction of Bus Station Network with Weights*

According to the above method, the weight of each edge, that is, the bus driving efficiency in the bus station network is calculated, and then the new adjacency matrix is constructed according to the weight.

## 4. Bus Network Efficiency and Network Optimization

*4.1. Static Bus Network Efficiency*

In [1], a method of network efficiency evaluation was proposed:

$$Eff(G) = \frac{1}{n(n-1)} \sum_{i \neq j \in G} e_{ij} = \frac{1}{n(n-1)} \sum_{i \neq j \in G} \frac{1}{d_{ij}} \tag{5}$$

$n$ is the number of nodes in the network; $d_{ij}$ is the shortest path length between any node $i$ and node $j$ in a network $G$, that is, the number of edges on the shortest path between the two nodes. Obviously, the greater the $Eff(G)$, the higher the efficiency of the network, that is, the connection performance of any two nodes in the network is better.

This definition of network efficiency is general, and is suitable for simple topological structure networks or networks with different weights. $d_{ij}$ can be extended in the actual network, for example, in the geographical distance weighted urban public transport network model, the meaning of $d_{ij}$ is the geographical distance between station $i$ and station $j$.

*4.2. Improved Transit Network Efficiency*

The efficiency calculation method of the public transport network mentioned above has some limitations. In a specific network, if the weight has a more profound meaning, then the method is no longer applicable [15].

This paper proposes a bus network with weights, the weight of each edge is the bus driving efficiency of the adjacent stations on the same line. $W_{ij}$ represents the sum of the value of maximum

path efficiency and the value of bus driving efficiency between bus station $i$ and station $j$, and $W_{ij}$ is improved by the improved calculation method. The network efficiency calculation formula applicable for the urban public traffic weighted network model can be obtained:

$$Eff(G) = \frac{1}{n(n-1)} \sum_{i \neq j \in G} e_{ij} = \frac{1}{n(n-1)} \sum_{i \neq j \in G} W_{ij} \tag{6}$$

### 4.3. Bus Network Optimization Method

The population density of a city will change with time, and the population density will increase in some areas. When this occurs, it is necessary to adjust the urban public transport network, otherwise it will lead to road congestion; similarly, when the population density of a region decreases, adjusting the site distribution appropriately will save cost. Therefore, we need to optimize the bus network [16,17].

#### 4.3.1. Efficiency of Transit Network before Optimization

The initial efficiency of the bus network before optimization is as follows:

$$Eff_0(G) = \frac{1}{n(n-1)} \sum_{i \neq j \in G} e_{ij} = \frac{1}{n(n-1)} \sum_{i \neq j \in G} W_{ij} \tag{7}$$

This efficiency is the efficiency of the bus network with no optimized edge.

#### 4.3.2. Delete Bus Station

When the bus driving efficiency between the three connected stations $i$, $j$, $k$ is very low, or if after deleting the intermediate station $k$, the impact on the entire bus complex network will be very small, we can delete the bus station $k$ to reduce cost.

By traversing and deleting each existing station in the bus network, the change rate of the efficiency of the whole bus network can be calculated, and the stations with the lowest change rate of the bus network can be deleted. The calculation formula of the change rate of public transport network efficiency is:

$$\Delta D_i = \frac{Eff\left(G + E_{ij} - E_{ik} - E_{kj}\right) - Eff_0(G)}{Eff_0(G)}, \ i \in V, \ j \in V \tag{8}$$

The required station search method is as follows:

$$min\left\{ \frac{Eff\left(G + E_{ij} - E_{ik} - E_{kj}\right) - Eff_0(G)}{Eff_0(G)} \right\} \tag{9}$$

$\Delta D_i$ is the change rate of the efficiency of the bus network after deleting the station $k$; $Eff\left(G + E_{ij} - E_{ik} - E_{kj}\right)$ is the efficiency of the entire bus network after deleting the station $k$. $E_{ik}$ is the edge between the station $i$ and the station $k$; $E_{kj}$ is the edge between the station $k$ and the station $j$; $E_{ij}$ is the edge between the station $i$ and the station $j$.

When one station is deleted, the change rate of the whole bus network is the smallest, indicating that the station has the least influence on the whole network, so it can be deleted to reduce the cost.

Figure 4 is a method diagram for deleting a station, and Figure 5 is a flow chart for deleting a station.

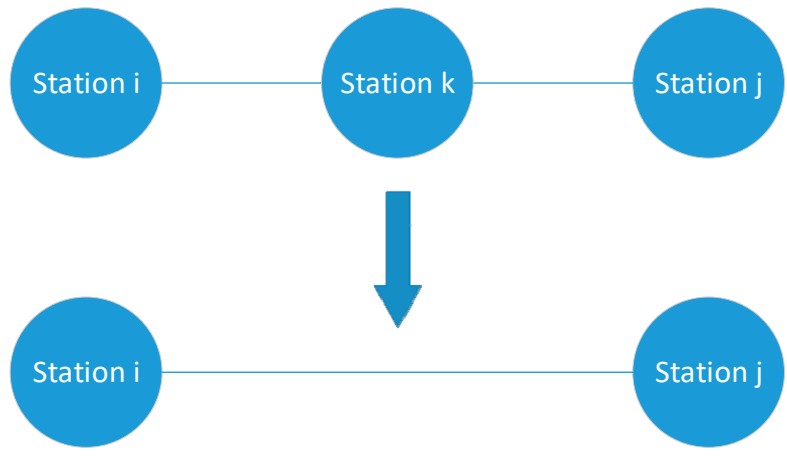

**Figure 4.** Delete bus station.

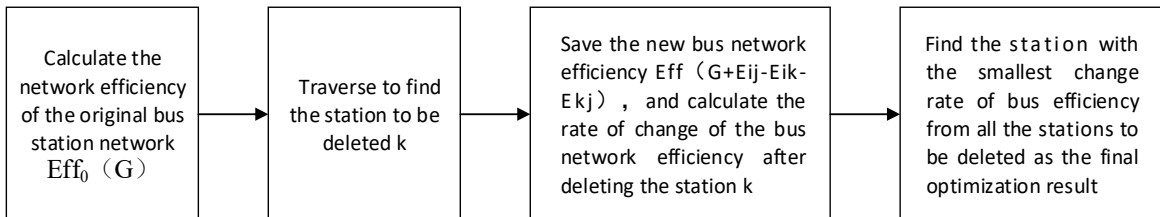

**Figure 5.** Delete bus stops flow chart.

### 4.3.3. Add Bus Station

It can be seen from Figure 3 that when the passenger load rate reaches a certain extent, if the passenger load rate between the station *i* and the station *j* continues to increase, it may cause road congestion, resulting in a decrease in the bus driving efficiency of the line. At this point, we need to add a new station *k* between station *i* and the station *j* to share the passenger flow.

Because this paper is based on the actual situation of Jinan city bus line operation for the purpose of analysis and calculation, there is no actual weight data in the new link of the bus network, so the actual weight after adding stations is not calculated.

However, the network efficiency after adding stations can be calculated. Assuming that a new station *k* is added, it is connected with the existing stations *i* and *j* in the bus network, and the change rate of network efficiency is:

$$\Delta A_i = \frac{Eff\left(G - E_{ij} + E_{ik} + E_{kj}\right) - Eff_0(G)}{Eff_0(G)}, \ i \in V, \ j \in V \tag{10}$$

The method for searching the connection point needed for optimization is:

$$max\left\{ \frac{Eff\left(G - E_{ij} + E_{ik} + E_{kj}\right) - Eff_0(G)}{Eff_0(G)} \right\} \tag{11}$$

$\Delta A_i$ is the change rate of the transit network efficiency after adding the station; $Eff\left(G - E_{ij} + E_{ik} + E_{kj}\right)$ is the efficiency of the entire bus network after adding the station *k*; $E_{ik}$ is the edge between the station *i* and the station *k*; $E_{kj}$ is the edge between the station *k* and the station *j*; $E_{ij}$ is the edge between the station *i* and the station *j* before the network optimization.

The location at which the change rate of the whole bus network is the largest is the location at which the station *k* will have the greatest effect on improving the whole bus network, so station *k* can be added.

Figure 6 is a method diagram for adding stations, and Figure 7 is a flow chart for adding stations.

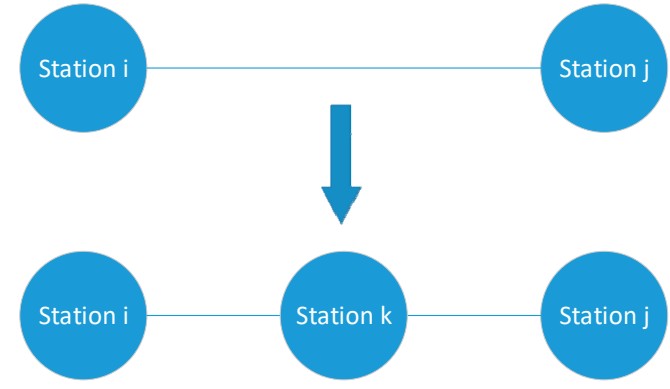

**Figure 6.** Add bus stations.

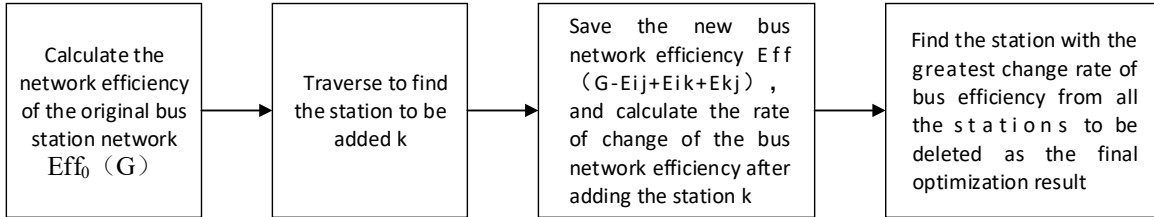

**Figure 7.** Add bus stops flow chart.

## 5. Simulation Results

Because the number of real bus network stations is large and the connection is complex, it is very difficult to obtain the real traffic flow, bus waiting time, number of travelers and other data of all bus stations and their sections. This paper constructs a small connected network using three bus lines to verify the feasibility of the method; it contains 30 bus stations, with bus line 1 containing 10 stations, bus line 2 containing 11 stations, and bus line 3 containing 12 stations. These three lines are intertwined and randomly weighted for each connecting edge within a reasonable range, as shown in Figure 8. Figure 9 shows part of the weighted adjacency matrix (30 * 30) of the undirected network shown in Figure 8. The matrix is symmetrical. If the elements corresponding to a row and a column in the matrix are not zero, it means that the two stations belong to the adjacent stations on the same line, that is, they are directly connected. If the elements corresponding to a row and a column in the matrix are zero, it means that there is no direct connection between the two stations. The initial network efficiency of the simulation bus network is 0.0711 by iterative calculation of the global efficiency of the network.

### 5.1. Analysis of Delete Bus Station Results

In the process of deleting a station, the weight of the newly generated connection edge of the direct connection station pair is equal to the weight of the two small edges before deletion. Finally, the results of the simulation experiment are as follows: delete station number 13 and connect station number 12 directly with station number 14. After deleting station 13, the efficiency of the new bus network is 0.0654, which is the smallest change in the efficiency of the bus network. The bus network diagram after deleting the station is shown in Figure 10.

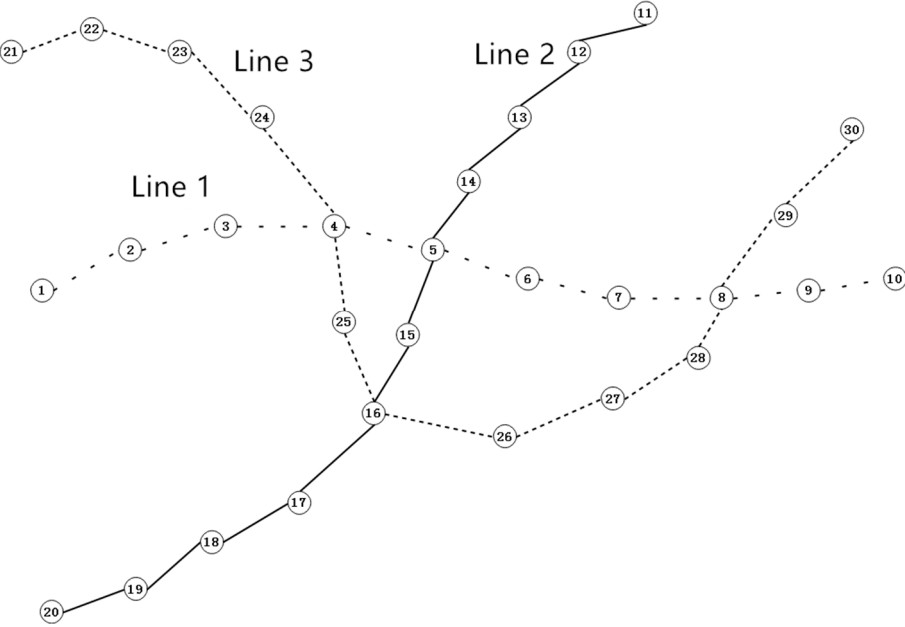

**Figure 8.** Simulated bus network connection diagram.

```
0 1.04170801720661  0 0 0 0 0 0 0 0 0 0 0 0 0 0 0 0 0 0 0 0 0 0 0 0 0 0 0 0
1.04170801720661  0 7.97419418240352  0 0 0 0 0 0 0 0 0 0 0 0 0 0 0 0 0 0 0 0 0 0 0 0 0 0 0
0 7.97419418240352  0 8.35572898588090  0 0 0 0 0 0 0 0 0 0 0 0 0 0 0 0 0 0 0 0 0 0 0 0 0 0
0 0 8.35572898588090  0 8.81825234827159  0 0 0 0 0 0 0 0 0 0 0 0 0 0 0 0 0 0 0 0 0 0 6.598496183365
0 0 0 8.81825234827159  0 1.75992260959819  0 0 0 0 0 0 0 2.30459318401354  8.67728005949704
0 0 0 0 1.75992260959819  0 4.59804384189007  0 0 0 0 0 0 0 0 0 0 0 0 0 0 0 0 0 0 0 0 0 0
0 0 0 0 0 4.59804384189007  0 3.33883362565589  0 0 0 0 0 0 0 0 0 0 0 0 0 0 0 0 0 0 0 0 0
0 0 0 0 0 0 3.33883362565589  0 8.20061632201877  0 0 0 0 0 0 0 0 0 0 0 0 0 0 0 0 0 0 0 1.6837
0 0 0 0 0 0 0 8.20061632201877  0 4.88272444717190  0 0 0 0 0 0 0 0 0 0 0 0 0 0 0 0 0 0 0
0 0 0 0 0 0 0 0 4.88272444717190  0 0 0 0 0 0 0 0 0 0 0 0 0 0 0 0 0 0 0 0 0
0 0 0 0 0 0 0 0 0 0 9.19582834986571  0 0 0 0 0 0 0 0 0 0 0 0 0 0 0 0 0 0 0
0 0 0 0 0 0 0 0 0 9.19582834986571  0 2.63662325472567  0 0 0 0 0 0 0 0 0 0 0 0 0 0 0 0 0
0 0 0 0 0 0 0 0 0 0 2.63662325472567  0 3.37422624869791  0 0 0 0 0 0 0 0 0 0 0 0 0 0 0
0 0 0 0 2.30459318401354  0 0 0 0 0 0 3.37422624869791  0 0 0 0 0 0 0 0 0 0 0 0 0 0 0 0 0
0 0 0 0 8.67728005949704  0 0 0 0 0 0 0 0 2.30985082346245  0 0 0 0 0 0 0 0 0 0 0 0 0 0
0 0 0 0 0 0 0 0 0 0 0 0 0 2.30985082346245  0 2.22461702837797  0 0 0 0 0 0 5.6192458588
0 0 0 0 0 0 0 0 0 0 0 0 0 0 2.22461702837797  0 8.82362986876080  0 0 0 0 0 0 0 0 0 0 0
0 0 0 0 0 0 0 0 0 0 0 0 0 0 0 8.82362986876080  0 6.21734128629013  0 0 0 0 0 0 0 0 0 0
0 0 0 0 0 0 0 0 0 0 0 0 0 0 0 0 6.21734128629013  0 5.94874181652699  0 0 0 0 0 0 0 0 0
0 0 0 0 0 0 0 0 0 0 0 0 0 0 0 0 0 5.94874181652699  0 0 0 0 0 0 0 0 0 0
0 0 0 0 0 0 0 0 0 0 0 0 0 0 0 0 0 0 2.10987041351649  0 0 0 0 0 0 0
0 0 0 0 0 0 0 0 0 0 0 0 0 0 0 0 0 0 2.10987041351649  0 2.65517009454175  0 0 0 0 0 0 0
0 0 0 0 0 0 0 0 0 0 0 0 0 0 0 0 0 0 2.65517009454175  0 3.15957273098413  0 0 0 0 0 0
0 0 0 6.59849618336559  0 0 0 0 0 0 0 0 0 0 0 0 0 3.15957273098413  0 0 0 0 0 0
0 0 0 4.15857142803044  0 0 0 0 0 0 0 0 0 5.61924585880348  0 0 0 0 0 0 0 0 0 0 0
0 0 0 0 0 0 0 0 0 0 0 0 0 4.61627230376748  0 0 0 0 0 0 0 0 0 4.75540362175933  0 0 0
0 0 0 0 0 0 0 0 0 0 0 0 0 0 0 0 0 0 0 0 0 0 4.75540362175933  0 1.44688987293168  0 0
0 0 0 0 0 0 0 1.68370022521758  0 0 0 0 0 0 0 0 0 0 0 0 0 0 0 0 0 1.44688987293168  0 0 0
0 0 0 0 0 0 0 3.15924538198292  0 0 0 0 0 0 0 0 0 0 0 0 0 0 0 0 0 0 0 0 9.12444498923753
0 0 0 0 0 0 0 0 0 0 0 0 0 0 0 0 0 0 0 0 0 0 0 0 0 0 0 0 9.12444498923753  0
```

**Figure 9.** Part of simulated public transport network authorized adjacency matrix.

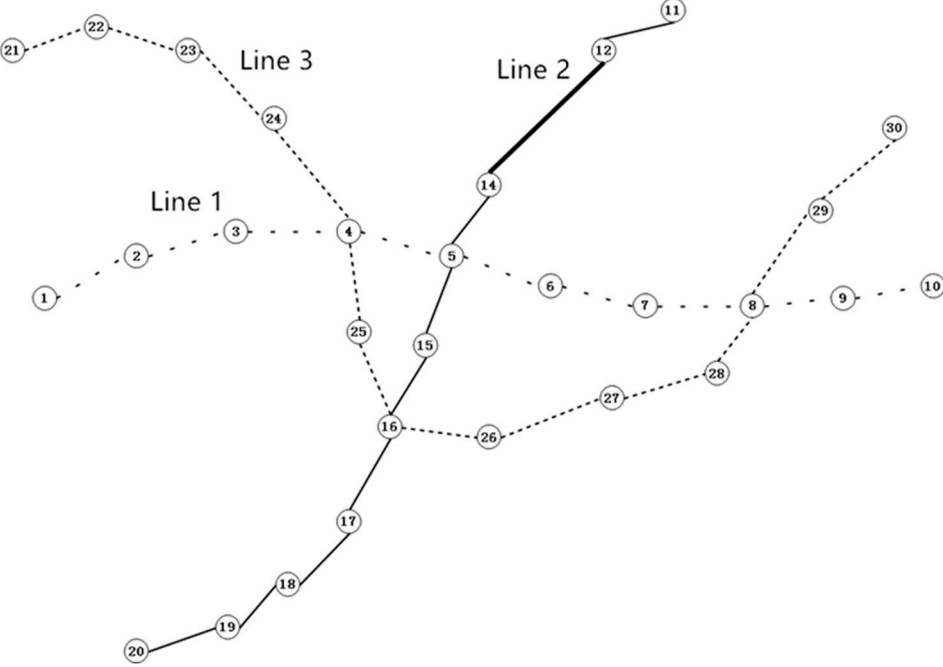

**Figure 10.** Bus network diagram after deleting the bus station.

*5.2. Analysis of Add Bus Station Results*

In the process of adding stations, this paper sets the weights of the newly generated two new links at one-half of the weights of the long ones that have not been added before. Finally, the results of adding bus stations in this simulation experiment are as follows: adding a new station 31 between the station numbered 26 and the station numbered 27; adding the new station to the bus network alters the efficiency to 0.2837, which is the greatest change in bus network efficiency. The bus network diagram after adding the new station is shown in Figure 11.

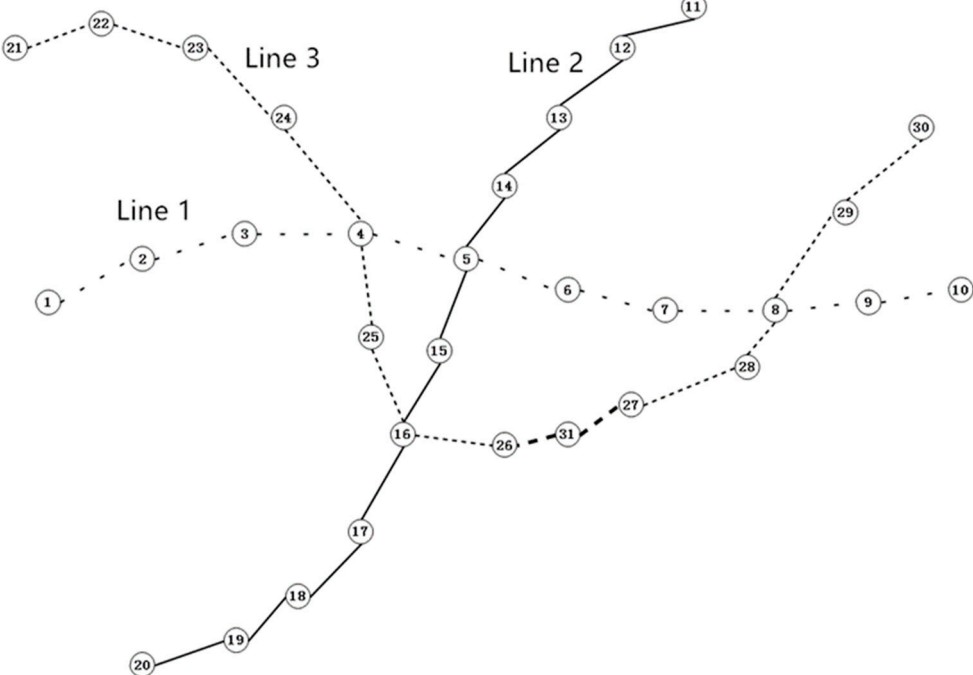

**Figure 11.** Bus network diagram after adding the new station.

## 6. Summary

With the rapid expansion of city scale and population structure, public transport has become an effective means to solve the problem of inconvenience for residents and traffic congestion. At this stage, the number of bus operation system stations is numerous, the line structure is complex, and the choice of bus routes is large. Based on theories of complex networks, graph theory, and intelligent transportation, this paper focuses on the modeling of urban public transport network, the definition and improvement of the weights of bus station network, and the optimization of the bus network structure. The main innovations and achievements of this paper are as follows:

With respect to the modeling method for public transport networks and the method of defining weights for public transport networks, by comparing the research work and progress of domestic and foreign scholars on complex public transport networks, and combining this with the existing stations in Jinan City, a complex network diagram of the public transport stations in Jinan City was constructed. At the same time, a method for defining weights was proposed. It uses not only distance or time or bus passenger flow as the connecting weights of bus station network, but also an improved BPR function for the connecting weights of the bus station network, which improves the comprehensiveness and accuracy of the connecting weights. In response to the shortcomings of the traditional public transport network efficiency algorithm, an improved network efficiency calculation method as proposed. This method combines graph theory with factors such as bus waiting time, passenger carrying rate and bus passenger flow. On the basis of the weighted public transport network, the bus driving efficiency, which can objectively reflect the actual road conditions, is used as the weight of the connecting edges in the network. At the same time, two network optimization methods, deleting sites and adding sites, were proposed to optimize the network. This method can not only be used to improve the network of bus stations, but can also play a role in the field of taxi operation, and can be used in the planning of urban taxi reception sites.

Although this paper has made some efforts in the optimization of traffic network, there are still many problems that need to be further studied, such as the lack of comprehensive consideration of the weight definition and the lack of consideration of the impact of optimization methods on specific routes. Therefore, we need to consider more complex factors in actual road conditions for example analysis, in order to make the bus network optimization more reasonable.

**Author Contributions:** P.D., D.L. and J.X. conceived and designed the study. P.D., D.L. performed the experiments. J.X., H.D., Y.W. provided the data. P.D. and D.L. analyzed the data. P.D. and D.L. proved the algorithm, P.D. wrote the paper. All authors read and approved the paper.

**Acknowledgments:** This work was supported by China Computer Program for Education and Scientific Research (NGII20161001), CERENT Innovation Project (NGII20170101), Shandong Province Science and Technology Projects (2017CXGC0202-2).

**Conflicts of Interest:** The authors declare no conflict of interest.

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
