# Peer review of "A Method of Bus Network Optimization Based on Complex Network and Beidou Vehicle Location"

_futureinternet, doi:10.3390/fi11040097_

Round 1

Reviewer 1 Report

In "A Method of Bus Network Optimization Based on Complex Network and Beidou Vehicle Location" authors study an important problem, and results are presented that perhaps might merit publication. The following comments needs to be taken into account if a revision will be granted.

1) With all due respect to authors that write technical text in a non-native language, there are nevertheless certain minimal standards that have to be met. This work fails in this respect, and therefore the English style and grammar should be improved as much as possible. I would recommend asking a colleague for help.

2) It would also improve the paper if the table and figure captions would be made more self-contained. In addition to briefly stating what is shown, one could also consider a sentence or two saying what is the main message of each table and figure. 

3) Some references contain errors, missing or incorrect information, and inconsistent formatting. It is difficult to give credit to research if such elementary aspects of the work are not error free. References should thus be corrected with the best care.

4) The results section is quite short, and I wonder whether the authors could be more specific in terms of description. Are all the details provided to allow others to reproduce the approach?

5) In the discussion, is it possible to list other applications that may benefit from the same combination of techniques?

6) Figures are of very poor quality, and in particular the adjacency matrix are not the way this should be presented. A color map, at least, is needed. Figure 1 is also not clear at all. How can it be that it is double diagonal? Why show this in the first place?

7) The introduction is also very short and modest, and it does not give due credit to recent research along similar lines. Good reviews are Information cascades in complex networks, J. Complex Netw. 5, 665-693 (2017) and Saving human lives: What complexity science and information systems can contribute, J. Stat. Phys. 158, 735-781 (2015).

If a revision will be granted, I will be happy to review this manuscript again.

Author Response

Reviewer 1's comments:(1)With all due respect to authors that write technical text in a non-native language, there are nevertheless certain minimal standards that have to be met. This work fails in this respect, and therefore the English style and grammar should be improved as much as possible. I would recommend asking a colleague for help.

Our response: I have made some changes to the English grammar.Thank you very much for your comments.

(2) It would also improve the paper if the table and figure captions would be made more self-contained. In addition to briefly stating what is shown, one could also consider a sentence or two saying what is the main message of each table and figure.

Our response:I added introductory text to some of the pictures in the article.

(3)Some references contain errors, missing or incorrect information, and inconsistent formatting. It is difficult to give credit to research if such elementary aspects of the work are not error free. References should thus be corrected with the best care.

Our response:I have made some modifications to the citation of references.

(4)The results section is quite short, and I wonder whether the authors could be more specific in terms of description. Are all the details provided to allow others to reproduce the approach?

Our response:As for the results of this article, I have revised and filled in a lot of content.

(5)In the discussion, is it possible to list other applications that may benefit from the same combination of techniques?

Our response:At the end of the article, I list the areas of application that might benefit.

(6)Figures are of very poor quality, and in particular the adjacency matrix are not the way this should be presented. A color map, at least, is needed. Figure 1 is also not clear at all. How can it be that it is double diagonal? Why show this in the first place?

Our response:In the matrix, weightless (0, 1) is used to represent the connection relationship between bus stops.At the same time, I have made corresponding amendments to this part of the article.

(7)The introduction is also very short and modest, and it does not give due credit to recent research along similar lines. Good reviews are Information cascades in complex networks, J. Complex Netw. 5, 665-693 (2017) and Saving human lives: What complexity science and information systems can contribute, J. Stat. Phys. 158, 735-781 (2015).

Our response:About the introduction of the article, I made some modifications and filled in some content.Thank you very much for your comments on my article, which has benefited me a lot.

Reviewer 2 Report

The authors model a bus network to assess optimal number of bus stops to maximize vehicle efficiency. The paper is written moderately well and seems easy enough to follow along with what they did in most places. I do have some concerns about omissions in their model. Perhaps these can be resolved.

What the authors don't seem to include in their work is dwell times, and the effect that bus stop changes will have on the amount of time it takes to load and unload people. The network optimization presented is one that optimizes vehicular efficiency. This is but one way to evaluate a transit network, and may not be the ideal way to evaluate. 

Another omission is transfers, which should include headways. These should be included in Section 3.3. The network is optimized without concern for operational characteristics, which is problematic for practical applications. Stations that facilitate transfers should be more valuable to the overall efficiency of the network than stations that do not facilitate transfers. 

I don't think the weighting described in Section 3 of page 2 captures all the characteristics claimed, but perhaps I misunderstand what the weighting does. If this is the case--and entirely possible, then my above comments are somewhat resolved, but not completely. As written, the weights are 0 or 1 based on adjacency. Then the authors discuss other characteristics. 

Lastly, there is no consideration of where the buses go, such as to business districts. This is a puzzling omission. A more central node matters more than a node on the circumference of the network. This isn't captured through adjacency.

I'm not sure what Figure 3 shows. It is not an "intuitive complex network diagram." It is fuzzy and doesn't convey what the authors want it to convey. I suggest a new, clearer figure, as well as better text describing what the reader should take away from the figure. 

Author Response

(1)What the authors don't seem to include in their work is dwell times, and the effect that bus stop changes will have on the amount of time it takes to load and unload people. The network optimization presented is one that optimizes vehicular efficiency. This is but one way to evaluate a transit network, and may not be the ideal way to evaluate.

Another omission is transfers, which should include headways. These should be included in Section 3.3. The network is optimized without concern for operational characteristics, which is problematic for practical applications. Stations that
facilitate transfers should be more valuable to the overall efficiency of the network than stations that do not facilitate transfers.

Our response:Residence time and transfer time are included in BPR function, and the BPR function values of different sections are different.

(2)I don't think the weighting described in Section 3 of page 2 captures all the characteristics claimed, but perhaps I misunderstand what the weighting does. If this is the case--and entirely possible, then my above comments are somewhat resolved, but not completely. As written, the weights are 0 or 1 based on adjacency. Then the authors discuss other characteristics.

Our response:In this paper, the driving efficiency of public transport is used to represent the weight and various factors are taken into account.

(3)there is no consideration of where the buses go, such as to business districts. This is a puzzling omission. A more central node matters more than a node on the circumference of the network. This isn't captured through adjacency.

Our response:On the issue of business districts: We differentiate congested and unobstructed sections according to their weights

(4)I'm not sure what Figure 3 shows. It is not an "intuitive complex network diagram." It is fuzzy and doesn't convey what the authors want it to convey. I suggest a new, clearer figure, as well as better text describing what the reader should take away from the figure.

Our response:For Figure 3, I have made some modifications in the article, hoping that the improved content will make the reader understand the role of Figure 3 more clearly.

Thank you very much for your comments on my article, which has benefited me a lot.

Round 2

Reviewer 1 Report

The authors have revised their manuscript comprehensively and with love to detail. I warmly recommend publication in present form.